# The SARM1 TIR domain produces glycocyclic ADPR molecules as minor products

Jeremy Garb[1☉], Gil Amitai[1☉], Allen Lu[2,3], Gal Ofir[1¤], Alexander Brandis[4], Tevie Mehlman[4], Philip J. Kranzusch[2,3,5], Rotem Sorek[1]*

1 Department of Molecular Genetics, Weizmann Institute of Science, Rehovot, Israel, 2 Department of Microbiology, Harvard Medical School, Boston, MA, United States of America, 3 Department of Cancer Immunology and Virology, Dana-Farber Cancer Institute, Boston, MA, United States of America, 4 Life Sciences Core Facilities, Weizmann Institute of Science, Rehovot, Israel, 5 Parker Institute for Cancer Immunotherapy at Dana-Farber Cancer Institute, Boston, MA, United States of America

☉ These authors contributed equally to this work.
¤ Current address: Department of Molecular Biology, Max Planck institute for Biology Tübingen, Tübingen, Germany
* rotem.sorek@weizmann.ac.il

**Data Availability Statement:** All relevant data are within the manuscript and its Supporting Information files.

**Funding:** We thank A. Yaron, O. Abraham, N. Pursotham, and S. Hobbs for fruitful discussions

## Abstract

Sterile alpha and TIR motif-containing 1 (SARM1) is a protein involved in programmed death of injured axons. Following axon injury or a drug-induced insult, the TIR domain of SARM1 degrades the essential molecule nicotinamide adenine dinucleotide ($NAD^+$), leading to a form of axonal death called Wallerian degeneration. Degradation of $NAD^+$ by SARM1 is essential for the Wallerian degeneration process, but accumulating evidence suggest that other activities of SARM1, beyond the mere degradation of $NAD^+$, may be necessary for programmed axonal death. In this study we show that the TIR domains of both human and fruit fly SARM1 produce 1″–2′ and 1″–3′ glycocyclic ADP-ribose (gcADPR) molecules as minor products. As previously reported, we observed that SARM1 TIR domains mostly convert $NAD^+$ to ADPR (for human SARM1) or cADPR (in the case of SARM1 from *Drosophila melanogaster*). However, we now show that human and *Drosophila* SARM1 additionally convert ~0.1–0.5% of $NAD^+$ into gcADPR molecules. We find that SARM1 TIR domains produce gcADPR molecules both when purified *in vitro* and when expressed in bacterial cells. Given that gcADPR is a second messenger involved in programmed cell death in bacteria and likely in plants, we propose that gcADPR may play a role in SARM1-induced programmed axonal death in animals.

## Introduction

TIR (Toll/interleukin-1 receptor) domains are evolutionarily conserved protein domains that play key roles in innate immunity and cell-death pathways in animals, plants, and bacteria [1–4]. These domains frequently present catalytic activity targeting the molecule nicotinamide adenine dinucleotide ($NAD^+$) as a substrate [3, 5–9]. In both plants and bacteria, TIR domains were shown to cleave $NAD^+$ and process it into adenosine-containing molecules that act as

on SARM1 activity and biochemical analysis of gcADPR signaling, as well as the Sorek and Kranzusch lab members for comments on the manuscript. R.S. was supported, in part, by the European Research Council (grant no. ERC-AdG GA 101018520), Israel Science Foundation (MAPATS Grant 2720/22), the Deutsche Forschungsgemeinschaft (SPP 2330, Grant 464312965), the Ernest and Bonnie Beutler Research Program of Excellence in Genomic Medicine, Dr. Barry Sherman Institute for Medicinal Chemistry, Miel de Botton, the Andre Deloro Prize, and the Knell Family Center for Microbiology. P.J.K. was supported, in part, by the Pew Biomedical Scholars program and The Mathers Foundation. G. O. was supported by the SAERI doctoral fellowship. The funders had no role in study design, data collection and analysis, decision to publish, or preparation of the manuscript.

**Competing interests:** P.J.K, R.S, G.A and A.L. are inventors of a patent application related to the production and utility of gcADPR. R.S. is a scientific cofounder and advisor of BiomX and Ecophage. The rest of the authors declare no conflict of interest. This does not alter our adherence to PLOS ONE policies on sharing data and materials.

second messenger immune signals, activating programmed cell death in response to infection. For example, some plant TIR-containing immune receptors, once they recognize effectors of plant pathogens, generate phosphoribosyl adenosine monophosphate (pRib-AMP), ADP-ribose-ATP (ADPR-ATP), or di-ADPR molecules. These molecules bind and activate a complex involving the protein EDS1, triggering a signaling cascade that leads to plant resistance and cell death [7, 8]. Other plant TIR-containing immune proteins were shown to process $NAD^+$ into $1''-2'$ glycocyclic ADP ribose ($1''-2'$ gcADPR) and $1''-3'$ gcADPR molecules [5, 10, 11]. In bacteria, TIR domain proteins in an anti-phage system called Thoeris recognize phage infection, and then process $NAD^+$ into $1''-3'$ gcADPR. This molecule activates a protein called ThsA that leads to premature death of the infected bacterial cell prior to phage maturation [9, 10, 12]. In other bacterial immune systems such as CBASS, Pycsar and prokaryotic Argonaute systems, TIR domains serve as $NAD^+$-depleting factors [13–17]. These highly processive TIRs use their $NAD^+$-processing capacity to eliminate $NAD^+$ from the cell in response to phage infection, thus depleting the cell of energy and aborting phage infection [15, 17–19].

In human cells, TIR domains are frequently associated with Toll-like receptors and other immune adaptor proteins [1, 20]. These TIRs are considered catalytically inactive and they transfer the immune signal via protein-protein interactions [21–23]. However, there is one human TIR-domain containing protein, called sterile alpha and TIR motif-containing 1 (SARM1), in which the TIR domain is catalytically active [6]. SARM1 is a key player in a neuronal programmed axon death pathway called Wallerian degeneration, in which injured axons are degenerated in an orderly manner [24–26]. Wallerian degeneration is characterized by granular disintegration of the axonal cytoskeleton, mitochondrial swelling, and axon fragmentation [27]. The Wallerian degeneration pathway was shown to depend on the activation of SARM1 [28] which, once activated, cleaves $NAD^+$ and depletes it from the injured axon [6, 29, 30]. Indeed, cells in which SARM1 is mutated in the catalytic site do not undergo programmed axonal death following axonal insult [6].

While $NAD^+$ depletion by SARM1 is considered a key factor of Wallerian degeneration, the precise mechanism by which SARM1 activity causes Wallerian degeneration is not yet fully understood [25, 31, 32]. For example, the Axundead mutant in *Drosophila* is able to prevent axon degeneration even with SARM1 activation, indicating that there are additional factors downstream to SARM1-mediated $NAD^+$ depletion that are required for programmed axonal death [33]. Additionally, multiple studies showed that blocking $Ca^{2+}$ influx into axons can prevent axonal degeneration, suggesting a connection between SARM1 and $Ca^{2+}$ signaling [34, 35]. It was shown that neuronal depletion of $NAD^+$ by factors other than SARM1 does not induce Wallerian degeneration [36–38], and, accordingly, it was suggested that factors other than $NAD^+$ depletion by SARM1 participate in orchestrating Wallerian degeneration [25, 32]. In this study, we provide evidence that the TIR domain of SARM1 is capable of generating $1''-2'$ gcADPR and $1''-3'$ gcADPR molecules. We propose that these molecules might have a role as signaling molecules in the Wallerian degeneration pathway.

## Materials and methods

### Bacterial strains

For the generation of cell lysates, *E. coli* strain BL21(DE3) (Agilent) was grown in MMB (LB supplemented with 0.1 mM $MnCl_2$ and 5 mM $MgCl_2$) at 37˚C. Whenever applicable, media were supplemented with chloramphenicol (30 µg mL$^{-1}$) or kanamycin (50 µg mL$^{-1}$), to ensure the maintenance of plasmids. For protein purification, *E. coli* strain BL21(DE3) (Agilent) was grown in 2YT media (1.6% Bacto-tryptone, 1% yeast extract, 0.5% NaCl) at 37˚C in the presence of ampicillin (100 µg mL$^{-1}$) to ensure maintenance of plasmids.

## Plasmid and strain construction

The human SARM1 TIR and SARM1 TIR E642A used for the cell lysates experiments were synthesized with codon optimization and cloned into a pET28 backbone by Twist Bioscience (S1 and S2 Files). The *Drosophila* SARM1 TIR used for the cell lysates experiments was synthesized with codon optimization by Genscript Corp and then cloned by Gibson assembly into a pACYC backbone with a Twin Strep tag fused to the N-terminus (S3 File). The E919A mutation was introduced using a KLD Enzyme Mix (NEB, no. M0554) with primers CCAGTCCTTACAATCTTCGTC and GTACATCGGGcGATCGTAGCGG (S4 File). The human and *Drosophila* SARM1 TIR used for *in vitro* reactions were codon optimized for bacterial expression and cloned from synthetic DNA fragments (Integrated DNA Technologies) by Gibson assembly into a custom pET expression vector with an N-terminal 6× His tag and an ampicillin resistance gene (S5 and S6 Files). cmTad1, AbTIR$^{TIR}$, and ThsB' were cloned similarly into pET expression vectors with an N-terminal 6×His-SUMO tag (cmTad1, AbTIR$^{TIR}$) or a C-terminal 6×His tag (ThsB') as previously reported by Yirmiya et al., 2024. The human and *Drosophila* SARM1 TIR used in this paper are based on UniProt accessions Q6SZW1 and Q6IDD9, respectively. Only the TIR domain was used.

## SARM1 TIR domains expression and lysate preparation for LC-MS analyses

Overnight cultures of bacteria containing WT or mutated SARM1 TIR domains were diluted 1:100 in 200 mL MMB and incubated at 37°C with shaking (200 r.p.m.) until reaching OD$_{600}$ of 0.3. At this point, IPTG was added to a concentration of 1 mM and the temperature was dropped to 30°C for an additional 4 hours. 50 mL samples were collected and centrifuged at 4°C for 10 min to pellet the cells. The supernatant was discarded and the tube was frozen at −80°C. To extract the metabolites, 600 µl of 100 mM Na phosphate buffer at pH 8 was added to each pellet. The thawed samples were transferred to a FastPrep Lysing Matrix B 2 mL tube (MP Biomedicals catalogue no. 116911100) and lysed using FastPrep bead beater for 40 s at 6 m s$^{-1}$ in two rounds. Tubes were then centrifuged at 4°C for 15 min at 15,000$g$. Supernatant was transferred to Amicon Ultra-0.5 Centrifugal Filter Unit 3 kDa (Merck Millipore catalogue no. UFC500396) and centrifuged for 45 min at 4°C at 12,000$g$. Where specified, filtrate was incubated with cmTad1 as previously described in Leavitt et al., 2022. Filtrate was taken and used for ThsA activity assays and for liquid chromatography mass spectrometry (LC-MS) analyses.

## *In vitro* production and purification of gcADPR molecules

$1''$–$2'$ and $1''$–$3'$ gcADPR were produced and purified as previously described in detail [39]. In brief, purified recombinant AbTIR$^{TIR}$ and ThsB' were used to set up reactions with NAD$^+$. Reactions were carried out at room temperature for 24–48 hours before boiling at 95°C for 10–15 minutes. Samples were clarified by centrifugation (13,500 g, 20 min), passed through a 10 kDa filter, and cmTad1 was added to the filtrate. Mixtures were incubated at RT for 1 h to allow complex formation before washing by successive concentration and dilution in a 10 kDa concentration unit, first with PBS, then with water. Complexes were concentrated to >3 mM before boiling at 95°C for 10 minutes, centrifugation (13,500 g, 20 min), and filtering through a 3 kDa filter. For long-term storage and shipment, samples were vacuum dehydrated and kept at –20°C.

## ThsA NADase activity assay

The NADase reaction was performed in black 96-well half area plates (Corning, 3694). In each microwell, ThsA protein which was purified as previously described (Ofir et al., 2021), was

added to cell lysate or to a positive control of 100 mM sodium phosphate buffer pH 8 supplemented with 1″–3′ gcADPR standards. ThsA was added to a final concentration of 100 nM protein in a 50 μl final volume reaction. Five microlitres of 5 mM nicotinamide 1,N$^6$-ethenoadenine dinucleotide (εNAD, Sigma,N2630) solution was added to each well immediately before the beginning of measurements and mixed by pipetting to reach a concentration of 500 μM in the 50 μl final reaction volume. Plates were incubated inside a Tecan Infinite M200 plate reader at 25°C, and measurements were taken every 1 min at 300 nm excitation wavelength and 410 nm emission wavelength. Reaction rate was calculated from the linear part of the initial reaction.

## Quantification of metabolites by LC-MS/MS

Quantification of metabolites in cell lysates or in the *in vitro* reactions was carried out using an Acquity I-class UPLC system coupled to Xevo TQ-S triple quadrupole mass spectrometer (both Waters, US). The UPLC was performed using an Atlantis Premier BEH C18 AX column with the dimension of $2.1 \times 100$ mm and particle size of 1.7 μm (Waters). Mobile phase A was 20 mM ammonium formate at pH 3 and acetonitrile was mobile phase B. The flow rate was kept at 300 μl min$^{-1}$ consisting of a 2 min hold at 2% B, followed by linear gradient increase to 100% B during 5 min. The column temperature was set at 25°C and an injection volume of 1 μl. An electrospray ionization interface was used as ionization source. Analysis was performed in positive ionization mode. Metabolites were analyzed using multiple-reaction monitoring with argon as the collision gas, and detected based on retention times and MS/MS parameters of chemical standards (S1 Fig). Quantification was made using standard curve in 0–1 mM concentration range. $^{15}N_5$-adenosine 5′-monophosphate (Sigma) was added to standards and samples as internal standard (0.5 μM). TargetLynx (Waters) was used for data analysis.

## SARM1 TIR domain purification for *in vitro* reactions

Human and *Drosophila* SARM1 TIR expression plasmids were transformed into *E. coli* BL21 (DE3) (Agilent). Bacterial colonies were grown on LB agar plates, and 15 mL of 2YT media starter cultures were grown overnight at 37°C with 230 rpm shaking from three picked colonies. 1L of 2YT expression culture supplemented with 10 mM nicotinamide was seeded with 15 mL starter culture and grown at 37°C with 230 rpm shaking to an $OD_{600}$ of 2.5 before expression induction with 0.5 mM IPTG. The temperature was then lowered to 16°C, and cultures were harvested by centrifugation after 16–20 h. Cell pellets from 2 L of culture were resuspended in 120 mL lysis buffer (20 mM HEPES-KOH pH 7.5, 400 mM NaCl, 10% glycerol, 30 mM imidazole, 1 mM DTT), lysed by sonication, and clarified by centrifugation at 25,000g for 20 min. Lysate was passed over a gravity column of 8 mL of Ni-NTA resin (Qiagen), washed with 70 mL wash buffer (20 mM HEPES-KOH pH 7.5, 1M NaCl, 10% glycerol, 30 mM imidazole, 1 mM DTT), and eluted with 20 mL of lysis buffer supplemented to 300 mM imidazole. Eluate was dialyzed against storage buffer (20 mM HEPES-KOH pH 7.5, 250 mM KCl, 1 mM TCEP) overnight at 4°C, concentrated to >4 mg/mL, flash frozen, and stored at −80°C.

Protein purity was assayed by SDS-PAGE. ~2 μg purified protein was separated on a 15% bis-acrylamide SDS gel (S2 Fig), and sizes were estimated using Blue Protein Standard (New England Biolabs). N-terminally 6×His-tagged fusions of human and *Drosophila* SARM1 TIR have an expected molecular weight of 19.2 kDa.

## *In vitro* SARM1 TIR domain reactions and sample preparation

Purified human and *Drosophila* SARM1 TIR recombinant proteins were used to set up 0.3 mL reactions (10 μM protein, 1 mM NAD$^+$, 50 mM HEPES-KOH pH 7.5, 150 mM NaCl, 1 mM

TCEP). Reactions were carried out at 37°C for 16 h, stopped by filtering through a 3 kDa filter (Amicon), and stored at −20°C.

## Results

This study was initiated upon the surprising observation that lysates from cells expressing the TIR domain of SARM1 are able to activate the bacterial ThsA protein (Fig 1A, 1B and S1 Table). ThsA is an anti-phage enzyme specifically activated by the molecule $1''$–$3'$ gcADPR, but not by the canonical cADPR, ADPR, or other ADPR derivatives [5, 10, 12], and we therefore suspected that SARM1 activity might generate $1''$–$3'$ gcADPR. We cloned the TIR domain of SARM1 from both human and *Drosophila melanogaster* (fruit fly) and overexpressed these TIR domains within *Escherichia coli* cells. We then extracted cell lysates and filtered them through 3 kDa filters to retain only small molecules. The *in vitro* activity of ThsA was triggered by filtered lysates of both human and *D. melanogaster* SARM1 TIRs (Fig 1B). Lysates from bacteria expressing SARM1 TIR with mutations in the known catalytic glutamic acid residue did not induce the *in vitro* activity of ThsA (Fig 1B), suggesting that the enzymatic activity of SARM1 TIR is essential for producing $1''$–$3'$ gcADPR. In agreement with the well documented NADase activity of SARM1, cells expressing the SARM1 TIR domain were depleted of NAD$^+$ (Fig 1C).

To further examine whether the molecule present within SARM1 TIR-expressing cells was indeed gcADPR, we exposed the filtered cell lysates to Tad1, a phage-derived protein that is known to specifically bind and sequester $1''$–$3'$ gcADPR and $1''$–$2'$ gcADPR molecules, but not cADPR or ADPR [12]. Lysates from SARM1 TIR-expressing cells lost their ability to activate ThsA if pre-incubated with Tad1, suggesting that Tad1 had sequestered the $1''$–$3'$ gcADPR molecule from these lysates (Fig 1B).

To substantiate that SARM1 expression produces $1''$–$3'$ gcADPR, we subjected filtered cell lysates from *E. coli* cells overexpressing the human and *Drosophila* SARM1 TIR domains to targeted mass spectrometry analysis. Lysates from cells expressing both human and *Drosophila* SARM1 TIR domains showed a clear presence of $1''$–$3'$ gcADPR, while lysates of bacteria expressing catalytic site mutations in these domains did not (Fig 1D). These results suggest that when expressed in bacteria, the NADase activity of the SARM1 TIR domain generates $1''$–$3'$ gcADPR.

To test the extent to which SARM1 TIR domains produce $1''$–$3'$ gcADPR, we purified the TIR domains of SARM1 from human and *Drosophila* and incubated the purified proteins with NAD$^+$. As expected from previous studies [6, 40, 41], SARM1 TIR domains consumed NAD$^+$ efficiently. The NAD$^+$ molecules were largely converted into ADPR (in the case of human SARM1) or into cADPR (for *Drosophila* SARM1), as previously shown for these two proteins [6] (Fig 2). However, we were also able to detect the accumulation of $1''$–$3'$ gcADPR as well as the molecule $1''$–$2'$ gcADPR in both human and *Drosophila* samples. *In vitro*, the recombinant *Drosophila* SARM1 TIR produces about one $1''$–$3'$ gcADPR molecule for every 300 cADPR molecules it generates, and about 1:1000 $1''$–$2'$ gcADPR:cADPR molecules. The human SARM1 TIR seems to generate roughly 1 gcADPR molecule for every 200 conversions *in vitro*.

## Discussion

Our results demonstrate that the TIR domain of SARM1 generates $1''$–$2'$ and $1''$–$3'$ gcADPR molecules as minor products when expressed in or purified from bacteria. Generation of gcADPR molecules appears to be a conserved feature of SARM1 TIR activity, as this activity was observed for both human and insect SARM1 TIR domains. Although the majority of NAD$^+$ processing by the SARM1 TIR generates ADPR and cADPR molecules as products,

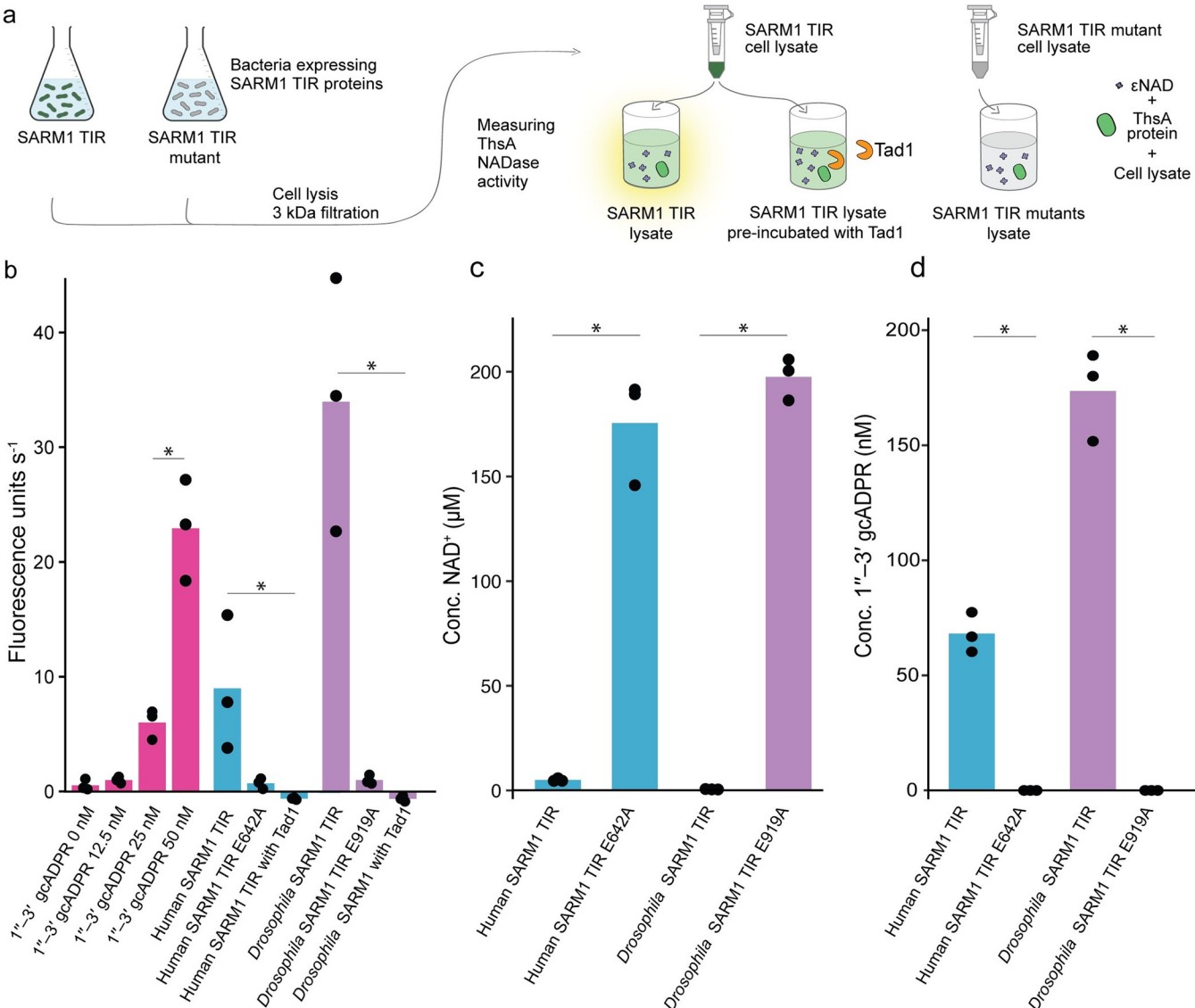

**Fig 1. Detection of 1″–3′ gcADPR in lysates of bacteria expressing the SARM1 TIR domain.** (a) Schematic representation of the experiment. Filtered lysates of cells expressing SARM1 TIR or SARM1 TIR active site mutants were tested for activation of the ThsA protein. NADase activity of ThsA was measured using a nicotinamide 1,N6-ethenoadenine dinucleotide (εNAD) cleavage fluorescence assay. (b) Activity of purified ThsA protein from *B. cereus*, incubated with increasing concentration of 1″–3′ gcADPR as well as with lysates derived from bacteria that express the SARM1 TIR domain from human and *D. melanogaster*. Data are also shown for lysates from bacteria expressing the SARM1 TIR domains with catalytic site mutations, as well as lysates that were pre-incubated with Tad1. Bars represent the mean of three independent replicates, with individual data points overlaid. Asterisks indicate a statistically significant difference (one-way ANOVA followed by pairwise multiple comparison analysis according to Tukey's honest significant difference criterion, $P < 0.05$). (c) LC-MS analysis showing concentrations of NAD$^+$ in cell lysates extracted from *E. coli* expressing human and *Drosophila* SARM1 TIR domains. Control cells in this experiment express SARM1 TIR domains with catalytic site mutations. Bar graphs represent the average of three independent replicates, with individual data points overlaid. Asterisk marks statistically significant increase (Student's *t*-test, two-sided, $P < 0.05$). (d) LC-MS analysis showing concentrations of 1″–3′ gcADPR in cell lysates extracted from *E. coli* expressing human and *Drosophila* SARM1 TIR domains. The cells used in this experiment are as in panel c. Bar graphs represent the average of three independent replicates, with individual data points overlaid. Asterisk marks statistically significant decrease (Student's *t*-test, two-sided, $P < 0.05$).

gcADPR molecules may accumulate in biologically meaningful concentrations in cells undergoing Wallerian degeneration. Human cells typically contain 0.2–0.5 mM of NAD$^+$ molecules [42], and if 0.1–0.5% of this NAD$^+$ is converted by SARM1 into gcADPR, this activity could generate high nanomolar to low micromolar levels of gcADPR in axons. We postulate that this

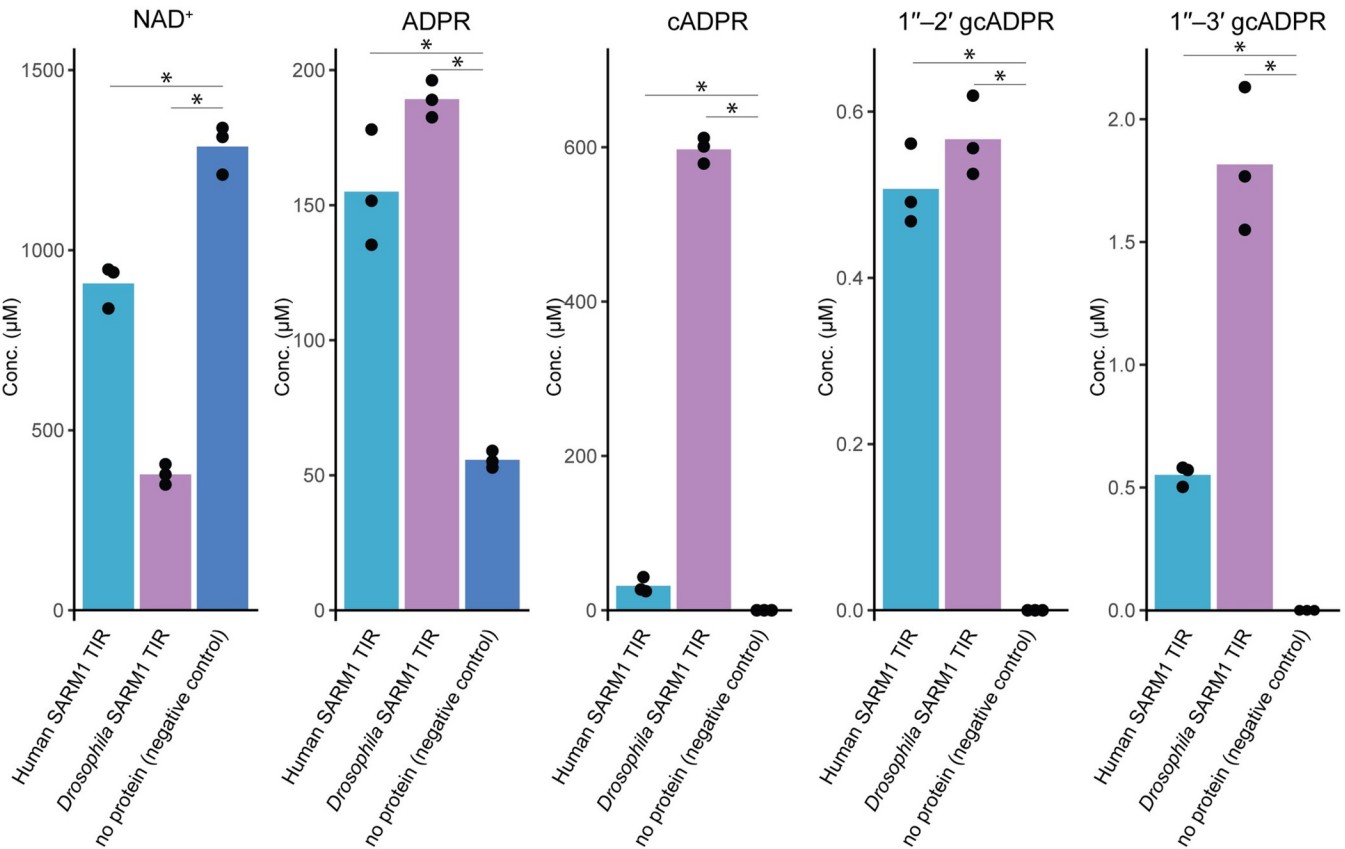

**Fig 2. LC-MS analysis of the products of *in vitro* reaction with purified SARM1 TIR domains.** Concentrations of molecules in filtered *in vitro* reactions containing purified human or *D. melanogaster* SARM1 TIR domains. 10 µM of protein was incubated with ~1 mM NAD$^+$ for 16 hours. Controls in this experiment had no protein added to the reaction. Bar graphs represent the average of three independent replicates, with individual data points overlaid. Asterisk marks statistically significant difference (Student's *t*-test, two-sided, $P < 0.05$).

molecule may function as a second messenger, in which case it would bind a receiver protein to trigger downstream activities contributing to programmed axonal death. Notably, intracellular second messenger signaling molecules, and specifically those involved in immunity and/ or the activation of cell death, typically activate their cognate receptor in high nanomolar to low micromolar concentrations [13, 15, 43]. An alternative explanation for our observations is that gcADPR production by SARM1 may be a nonfunctional byproduct of highly processive NADase activity of the SARM1 TIR domain. Notably, we observed production of gcADPR when SARM1 TIR domains were overexpressed in bacterial cells, and it is possible that the full length SARM1 protein does not generate these molecules when expressed at physiological levels in the native animal cells.

In bacteria, 1″–3′ gcADPR molecules are produced by TIR domains in response to phage infection. These molecules activate ThsA by specifically binding a domain called SLOG found at the C-terminus of ThsA. Binding of 1″–3′ gcADPR to the SLOG domain results in conformational changes that alter the oligomeric state of ThsA to activate the protein [9]. SLOG-like domains are also found in human proteins, specifically in calcium channels from the TRPM family where they were shown to bind ADPR derivatives [44, 45]. Given that activation of Ca$^{2+}$ influx has been linked to Wallerian degeneration downstream of SARM1 activity [34], it is worth understanding if calcium channel receptors could potentially respond to gcADPR produced by SARM1.

Although the NADase activity of SARM1 is well established as essential for Wallerian degeneration, the mechanism linking NAD$^+$ depletion to axonal degeneration is not yet entirely clear [25, 31, 32]. Indeed, previous studies have suggested that additional factors acting downstream to NAD$^+$ depletion by SARM1 may be necessary for the orderly death of injured axons [25, 33]. The gcADPR molecules revealed in this study as minor products of SARM1 TIR domains might be related to additional processes that could operate downstream of SARM1. Whether these molecules indeed accumulate in axons during Wallerian degeneration, and whether they have a biological function in axonal death, remains to be determined by future studies.

## Supporting information

**S1 Fig. Standards of NAD$^+$, ADPR, cADPR and gcADPR molecules as measured via targeted mass spectrometry.** Extracted mass chromatograms of specified ions, detected in a sample containing equal concentration of each standard, demonstrating the difference in retention times. Peak height is normalized to the highest peak in frame.
(TIF)

**S2 Fig. Protein purification of human and *Drosophila* SARM1 TIR proteins.** ~2 μg purified protein was separated on a 15% bis-acrylamide SDS gel and sizes were estimated using protein standard. N-terminally 6×His-tagged fusions of human and *Drosophila* SARM1 TIR have an expected molecular weight of 19 kDa (corresponding bands marked by asterisk).
(TIF)

**S1 Table. Raw data for Figs 1 and 2 (included as an excel sheet).**
(XLSX)

**S1 File. Plasmid map.**
(TXT)

**S2 File. Plasmid map.**
(TXT)

**S3 File. Plasmid map.**
(TXT)

**S4 File. Plasmid map.**
(TXT)

**S5 File. Plasmid map.**
(TXT)

**S6 File. Plasmid map.**
(TXT)

**S7 File. Raw gel images.**
(PDF)

## Acknowledgments

We thank A. Yaron, O. Abraham, N. Pursotham, and S. Hobbs for fruitful discussions on SARM1 activity and biochemical analysis of gcADPR signaling, as well as the Sorek and Kranzusch lab members for comments on the manuscript.

## Author Contributions

**Conceptualization:** Rotem Sorek.

**Data curation:** Alexander Brandis, Tevie Mehlman.

**Investigation:** Jeremy Garb, Gil Amitai, Allen Lu.

**Writing – original draft:** Rotem Sorek.

**Writing – review & editing:** Jeremy Garb, Gal Ofir, Philip J. Kranzusch.

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
