## [Decision Letter · Decision Letter 0]

6 Feb 2024

PONE-D-23-31863The SARM1 TIR domain produces glycocyclic ADPR molecules as minor productsPLOS ONE

Dear Dr. Garb,

Thank you for submitting your manuscript to PLOS ONE. After careful consideration, we feel that it has merit but does not fully meet PLOS ONE’s publication criteria as it currently stands. Therefore, we invite you to submit a revised version of the manuscript that addresses the points raised during the review process.

We look forward to receiving your revised manuscript.

Kind regards,

Asif Ali

Academic Editor

PLOS ONE

Journal Requirements:

   "We thank A. Yaron, O. Abraham, N. Pursotham, and S. Hobbs for fruitful discussions on SARM1 activity and biochemical analysis of gcADPR signaling, as well as the Sorek and Kranzusch lab members for comments on the manuscript. R.S. was supported, in part, by the European Research Council (grant no. ERC-AdG GA 101018520), Israel Science Foundation (MAPATS Grant 2720/22), the Deutsche Forschungsgemeinschaft (SPP 2330, Grant 464312965), the Ernest and Bonnie Beutler Research Program of Excellence in Genomic Medicine, Dr. Barry Sherman Institute for Medicinal Chemistry, Miel de Botton, the Andre Deloro Prize, and the Knell Family Center for Microbiology. P.J.K. was supported, in part, by the Pew Biomedical Scholars program and The Mathers Foundation. G.O. was supported by the SAERI doctoral fellowship."

   "P.J.K, R.S, G.A and A.L. are inventors of a patent application related to the production and utility of gcADPR. R.S. is a scientific cofounder and advisor of BiomX and Ecophage. The rest of the authors declare no conflict of interest."

Additional Editor Comments:

The manuscript titled "The SARM1 TIR domain produces glycocyclic ADPR molecules as minor products" by Garb et al. presents significant findings related to gcADPR molecules during Wallerian degeneration by the SARM1 TIR domain. Reviewer #1 highlights the need for statistical details, raw data in supplementary materials, and an exploration of evolutionary differences between Drosophila and human SARM1 TIR domains. Reviewer #2 expresses concern about the lack of evidence regarding the role of gcADPR molecules in inducing cell death, seeks an explanation for the Drosophila's higher catalytic activity, and recommends including stastical significance values for figures. Addressing these comments will enhance the manuscript's rigor and suitability for publication.

Reviewers' comments:

Reviewer's Responses to Questions

**Comments to the Author**

1. Is the manuscript technically sound, and do the data support the conclusions?

Reviewer #1: Yes

Reviewer #2: Yes

2. Has the statistical analysis been performed appropriately and rigorously? 

Reviewer #1: No

Reviewer #2: No

3. Have the authors made all data underlying the findings in their manuscript fully available?

Reviewer #1: No

Reviewer #2: Yes

4. Is the manuscript presented in an intelligible fashion and written in standard English?

Reviewer #1: Yes

Reviewer #2: Yes

5. Review Comments to the Author

Reviewer #1: Review Comments

The manuscript "The SARM1 TIR domain produces glycocyclic ADPR molecules as minor products" by Garb et al. is a good discovery about gcADPR molecules during Wallerian degeneration by SARM1 TIR domain. This research article provides importance of small molecules during stress conditions. This manuscript is of interest to the neurobiology and human biology researchers, as well as different biology researchers and I expect that the article will be well-cited. I have the following minor comments to consider.

1. Authors add statistical details in their figure’s legend and statistical section in materials and methods.

2. Authors should provide raw data of figures in supplementary with details?

3. In both figures, the Drosophila SARM1 TIR domain is more catalytically active than the human SARM1I, therefore I recommend that the authors consider your results from an evolutionary standpoint as well. Is there any difference in protein sequence in the Drosophila and human catalytic sites? Would like to give some insights about that in manuscript discussions?

Reviewer #2: The manuscript titled “The SARM1 TIR domain produces glycolytic ADPR molecules as minor products” discusses the generation of 1''-2 and 1''-3 glycolytic ribose molecules as subordinate products of TIR domains in both human and Drosophila SARM1. The authors, through well-defined experiments, have preliminarily concluded that these molecules are minor reaction products, yet they also suggest that these byproducts may play a role in SARM1-induced programmed axonal death in animals. However, they have not provided evidence for the latter part. Despite this, the novel findings are noteworthy and could be considered for acceptance once the following concerns are addressed.

1. In an invitro setting, it is unclear whether the 1”-2 and 1”-3 gcADPR molecules (in concentrations ranging from high nanomolar to low micromolar) alone or in combination with ADPR or cADPR induce cell death of cultured axons.

2. The inclusion of a mutated TIR domain protein in the mass spectrometry analysis presented in Figure 2 would enhance the study.

3. An explanation for the increased activity of the Drosophila TIR domain compared to the human TIR domain would be informative.

4. Please include the significance values of the figures.

6. PLOS authors have the option to publish the peer review history of their article (what does this mean?). If published, this will include your full peer review and any attached files.

Reviewer #1: **Yes: **Pawan Kumar

Reviewer #2: **Yes: **Suvranil Ghosh

While revising your submission, please upload your figure files to the Preflight Analysis and Conversion Engine (PACE) digital diagnostic tool, https://pacev2.apexcovantage.com/. PACE helps ensure that figures meet PLOS requirements. To use PACE, you must first register as a user. Registration is free. Then, login and navigate to the UPLOAD tab, where you will find detailed instructions on how to use the tool. If you encounter any issues or have any questions when using PACE, please email PLOS at <

---

## [Author Response · Author response to Decision Letter 0]

19 Mar 2024

A response to reviewers letter is included in this submission.

We thank the referees for their thoughtful comments and suggestions. Our point-to-point response to each of the referees’ comments is detailed below.

Reviewer #1: Review Comments

The manuscript "The SARM1 TIR domain produces glycocyclic ADPR molecules as minor products" by Garb et al. is a good discovery about gcADPR molecules during Wallerian degeneration by SARM1 TIR domain. This research article provides importance of small molecules during stress conditions. This manuscript is of interest to the neurobiology and human biology researchers, as well as different biology researchers and I expect that the article will be well-cited. 

Answer: Thank you for acknowledging the importance of our study!

I have the following minor comments to consider.

1. Authors add statistical details in their figure’s legend and statistical section in materials and methods.

Answer: We thank the reviewer for this valuable comment. We now added statistical details to figures, and explain the statistical analyses in the figure legends. 

2. Authors should provide raw data of figures in supplementary with details?

Answer: We thank the reviewer for this comment. We now included the raw data for figures 1 and 2 in a new Supplementary Table S1.

3. In both figures, the Drosophila SARM1 TIR domain is more catalytically active than the human SARM1I, therefore I recommend that the authors consider your results from an evolutionary standpoint as well. Is there any difference in protein sequence in the Drosophila and human catalytic sites? Would like to give some insights about that in manuscript discussions?

Answer: We thank the reviewer for this comment. We want to emphasize that the proteins were tested in conditions other than their natural conditions in neuronal cells – we either expressed them in bacteria (Figure 1B, 1C, 1D) or in vitro (Figure 2). For this reason, we would caution against interpretations regarding different enzymatic rates between the human and Drosophila SARM1 TIRs, due to the non-native environment. To clarify this point, we modified the Discussion text and it now reads: “Notably, we observed production of gcADPR when SARM1 TIR domains were overexpressed in bacterial cells, and it is possible that the full length SARM1 protein does not generate these molecules when expressed at physiological levels in the native animal cells.”

Reviewer #2: 

The manuscript titled “The SARM1 TIR domain produces glycolytic ADPR molecules as minor products” discusses the generation of 1''-2 and 1''-3 glycolytic ribose molecules as subordinate products of TIR domains in both human and Drosophila SARM1. The authors, through well-defined experiments, have preliminarily concluded that these molecules are minor reaction products, yet they also suggest that these byproducts may play a role in SARM1-induced programmed axonal death in animals. However, they have not provided evidence for the latter part. Despite this, the novel findings are noteworthy and could be considered for acceptance once the following concerns are addressed.

Answer: We thank the reviewer for acknowledging the novelty of our findings!

1. In an invitro setting, it is unclear whether the 1”-2 and 1”-3 gcADPR molecules (in concentrations ranging from high nanomolar to low micromolar) alone or in combination with ADPR or cADPR induce cell death of cultured axons.

Answer: as indicated by the reviewer in their summary above, in this study we did not test whether the molecules produced by SARM1 affect cell death in axons. We hope that our discoveries will prompt neurobiologists to examine this possibility, and hence our motivation in publishing the current manuscript. To make this point clear to the reader, our text in the Discussion section reads: “The gcADPR molecules revealed in this study as minor products of SARM1 TIR domains might be related to additional processes that could operate downstream of SARM1. Whether these molecules indeed accumulate in axons during Wallerian degeneration, and whether they have a biological function in axonal death, remains to be determined by future studies.”

2. The inclusion of a mutated TIR domain protein in the mass spectrometry analysis presented in Figure 2 would enhance the study.

Answer: We thank the reviewer for this comment, but we believe we addressed these mutations already in Figure 1. We performed our experiments including controls for active-site mutations when the SARM1 TIR was expressed in bacterial cells (Figure 1). The mutations included substitution of the active site glutamate in both the human and Drosophila SARM1 TIR, and in both cases the enzymatic activity was clearly and completely eliminated (Figure 1B, 1C, 1D). We feel that these mutations are already very convincing.

3. An explanation for the increased activity of the Drosophila TIR domain compared to the human TIR domain would be informative.

Answer: We thank the reviewer for this comment. We want to emphasize that the proteins were tested in conditions other than their natural conditions in human neuronal cells – we either expressed them in bacteria (Figure 1B, 1C, 1D) or in vitro (Figure 2). For this reason, we would caution against interpretations regarding different enzymatic rates between the human and Drosophila SARM1 TIRs, due to the non-native environment. To clarify this point, we modified the Discussion text and it now reads: “Notably, we observed production of gcADPR when SARM1 TIR domains were overexpressed in bacterial cells, and it is possible that the full length SARM1 protein does not generate these molecules when expressed at physiological levels in the native animal cells.”

4. Please include the significance values of the figures.

Answer: We thank the reviewer for this valuable comment. We have added statistical details to figures and explain the statistical analyses in the revised figure legends.

---

## [Editor Report · Decision Letter 1]

1 Apr 2024

The SARM1 TIR domain produces glycocyclic ADPR molecules as minor products

PONE-D-23-31863R1

Dear Dr. Garb,

We’re pleased to inform you that your manuscript has been judged scientifically suitable for publication and will be formally accepted for publication once it meets all outstanding technical requirements.

Kind regards,

Asif Ali

Academic Editor

PLOS ONE
---

## [Editor Report · Acceptance letter]

7 Apr 2024

PONE-D-23-31863R1 

PLOS ONE

Dear Dr. Garb, 

I'm pleased to inform you that your manuscript has been deemed suitable for publication in PLOS ONE. Congratulations! Your manuscript is now being handed over to our production team.

Kind regards, 

on behalf of

Dr. Asif Ali 

Academic Editor

PLOS ONE